# Effectiveness of Two Transcutaneous Electrical Nerve Stimulation (TENS) Protocols in Women with Provoked Vestibulodynia: A Randomized Controlled Trial

**DOI:** 10.3390/medsci11030048

**Published:** 2023-08-02

**Authors:** Filippo Murina, Dario Recalcati, Stefania Di Francesco, Irene Cetin

**Affiliations:** Lower Genital Tract Disease Unit, V. Buzzi Hospital–University of the Study of Milan, 20124 Milan, Italy

**Keywords:** TENS, dyspareunia, vestibulodynia, vulvar pain, vulvodynia, pelvic floor dysfunction

## Abstract

Background: Vestibulodynia (VBD) is the most common form of vulvodynia. Because VBD is a pain disorder, transcutaneous electrical nerve stimulation (TENS) can be used as treatment. This study aims to evaluate the effects of two-parameter combinations (frequency and pulse duration) of TENS in reducing pain intensity and dyspareunia in VBD. Methods: A randomized, double-blind, controlled trial was conducted to study the effect of two different electrical stimulation treatment regimens on women with VBD receiving domiciliary TENS. Outcomes were the mean change from baseline at 60 and 120 days of burning/pain and dyspareunia (VAS), Vulvar Pain Functional Questionnaire (V-Q), Female Sexual Functioning Index (FSFI) and vaginal electromyography measurements. Results: A total of 78 subjects, 39 in each group, completed the trial. Patients in Groups 1 and 2 received a mean of 46.9 and 48.4 TENS sessions. By day 120, there was a 38.2% reduction in the burning/pain and a 52.1% reduction in the dyspareunia VAS scores in Group 1, as compared to 21.3% (*p* = 0.003) and 23.1% in Group 2 (*p* = 0.01), respectively. FSFI, V-Q, and muscle-strength measures also improved but were not statistically significant. Conclusions: Our findings showed the potential of TENS in the treatment of VBD.

## 1. Introduction

A recent consensus statement has been made on the difference between persistent vulvar pain of at least three months triggered by a specific disease (e.g., inflammatory, neoplastic, traumatic) and vulvodynia, which is vulvar pain without an identifiable reason [1]. Vulvodynia occurs in approximately 8–15% of women during their lifetime [2,3], and new disease onset can be found in approximately 4.3 cases per 100 women per year [4]. Vulvodynia localized at the vestibule, vestibulodynia (VBD), is the most frequent manifestation of the disease (about 80%) and is characterized by tenderness or pressure in the vulvar vestibule [5]. Typically, pain is provoked by sexual intercourse, the use of tampons, and tight clothing. The etiology of VBD is multifactorial, and several pathophysiological mechanisms have been proposed, including inflammatory, hormonal, congenital, genetic, neuroproliferative, and muscular factors [5]. Hypersensitivity of the vulvar vestibule is a defining characteristic of VBD [6]. The pattern of VBD responses is suggestive of sensory abnormalities in the form of evoked pain, suggesting sensitization, an underlying manifestation of neuropathic pain. This is consistent with biopsies that showed increased innervation of the vulvar vestibule, subepithelial heparinase activity, and cytokines, which have been linked to a neuroinflammatory process [7].

This hyperinnervation correlates with allodynia (pain from light contact) and hyperalgesia (enhanced pain perception) [7,8]. Furthermore, VBD is often associated with pelvic floor muscle overactivity [9]. This chronic pattern can lead to decreased muscle perfusion, muscle hyperactivity, and the development of myofascial trigger points, resulting in localized pain or tearing and/or severe pain.

Transcutaneous electrical nerve stimulation (TENS) involves delivering an electrical current through the skin to treat pain and is effective in many clinical areas, including inflammatory, neurological, and non-inflammatory conditions [10]. Clinical studies in patients and animal models of neuropathic pain have shown that TENS reduces pain and allodynia [11]. The process of TENS-induced analgesia is thought to be multifactorial and likely encompasses peripheral, spinal, and supraspinal mechanisms [11]. The stimulation parameters (e.g., frequency, pulse duration, stimulation intensity) that are set on the TENS unit determine the type of nerve fibers stimulated and are important for its mechanism of action. Clinical studies evaluating the analgesic effects of TENS have focused on discussing different TENS parameters [11].

According to studies, TENS yielded a significant improvement in pain and sexual function in patients with VBD [12,13,14]. These findings suggest that TENS can be a potential component of VBD treatment. However, there are some biases on the wide variation in electrical stimulation parameters adopted and the duration of treatment required to obtain positive results. This study aims to evaluate the effects of the combination of two parameters (frequency and pulse duration) in reducing pain intensity and dyspareunia related to VBD, as well as the change in the pattern of symptoms relative to the number of TENS applied.

## 2. Materials and Methods

This was a randomized, single-blind, controlled trial performed between January 2019 and December 2019. Institutional Review Board approval for the study was obtained and registered at http://www.isrctn.com (ISRCTN12345678, accessed on 1 March 2021). Subjects were eligible for enrollment if they were at least 18 years of age and were not yet in menopause (absence of menstruation for 12 months), were diagnosed with VBD for at least six months, and underwent a standardized gynecological examination by one of our staff gynecologists.

The exclusion criteria were an active vulvovaginal infection at the time of their gynecological examination, genital bleeding of unknown origin, a cardiac pacemaker, or pregnancy. Patients enrolled in the trial signed an informed consent form and were randomly and blindly assigned to one of two groups with different parameter combinations of TENS therapy in a self-administered domiciliary protocol. Before randomization, patients were asked to stop any topical, systemic, or physical therapy at least 30 days before starting TENS therapy. Women with VBD completed questionnaires and underwent a gynecological examination, including a vulvo- and colposcopy. The International Society for the Study of Vulvovaginal Disease vulvodynia questionnaire was used in this study [15]. Women who were suitable for the study underwent a vestibular cotton swab examination (at the 1, 3, 5, 7, 9, and 11 o’clock positions around the vaginal opening at the Hart’s line) to confirm the diagnosis of VBD. The sensitivity around the vestibule was scored. At the first assessment, symptoms of burning/pain and dyspareunia were evaluated using a 10-cm visual analogue scale (VAS). Moreover, an Italian translation of the validated Vulvar Pain Functional Questionnaire (V-Q) [16] and the Female Sexual Functioning Index (FSFI) [17] were measured before and after treatment. The V-Q is an 11-item questionnaire provided to help patients quantify the extent in which vulvar pain is affecting their lives (i.e., “I can’t wear tight-fitting clothing”, “Gets worse when I walk or when I sit”, “I cannot use tampons at all”). The higher the score, the greater the functional limitation. The FSFI is a versatile self-report tool that measures the dimensions of female sexual function and consists of 19 questions divided into six domains: desire, arousal, lubrication, orgasm, satisfaction, and pain [17].

All subjects underwent vaginal surface electromyography (EMG). Vaginal EMG measurements were taken at rest and during several exercises of the pelvic floor, through an EMG device with a vaginal sensor (Myotonus plus©-London-UK). Pelvic floor muscle (PFM) activity at rest was measured as the mean muscle tone value at rest after six maximum contractions separated by a rest period of at least 12 s, while the PFM peak activity was calculated as the mean of six maximum voluntary contractions separated by a rest period of at least 12 s. The assessments also included the PFM strength obtained by subtracting the maximal value from the resting values [18].

The primary efficacy endpoint for this study included the mean change from baseline to 60 and 120 days of pain and dyspareunia. The secondary endpoints were variations in the vestibular cotton swab sensitivity and vaginal EMG measurements.

A dual-channel portable TENS unit (EVA©; Sirval, Milan, Italy) was used. The stimulation was delivered through a plastic vaginal probe connected to the device (Periprobe VAG2ST; Sirval, Milano, Italy) that was 20 mm in diameter and 110 mm in length with two gold metallic transversal rings as electrodes. The tube was inserted 20 mm into the vagina. The participants were randomized into two groups with different electrical stimulation treatment regimens. Two customized programs were set for each group according to our previous studies [12,13] and several trials on the use of TENS in neuropathic pain [11].

The electric parameters used in both treatments were:Group 1: 15 min of 100 Hz frequency, a pulse width of 50 µs, and time on:off 20:10 s (first program) followed by 15 min of 5 Hz frequency, a pulse width of 100 µs, and time on:off 20:10 s (second program).Group 2: 15 min of 60 Hz frequency, a pulse width of 50 µs, and time on:off 20:10 s (first program) followed by 15 min of 5 Hz frequency, a pulse width of 200 µs, and time on:off 20:10 s (second program).

Each patient underwent two to three assessments designed to familiarize them with the use of TENS while allowing the therapist to check if the patient was using the device properly. In the TENS treatment protocol, the pulse was increased rapidly until the patient reported any sensation under the electrodes. The intensity was then increased slowly until this sensation reached a level described as the maximum tolerable sensation without experiencing pain. After completing the trial, the patient obtained their TENS unit after verbal and written instructions, with a recommendation to perform home treatment three times per week.

Randomization was generated by a computer program, and the concealed allocation was achieved by varying the number of self-adhesive labels attached to the TENS unit according to the information provided in the envelope sent to the company representative. The envelope sent to the clinic contained information on which of the number-coded units should be handed out to the participants of the first and second groups. The data were all blinded from the commencement of the analysis until after the completion of study protocols.

The sample size was calculated based on averages and standard deviations of a mean pain reduction of 3.5 points based on the results of available studies regarding the use of TENS in a woman with VBD [12,13,14]. Based on the anticipated cure rate in one of the two study groups, a minimum of 35 subjects per group was required to detect an increase in the cure rate of at least a mean pain reduction of 4 points, assessed at the two-sided 5% level of significance with 80% power. Forty subjects per group were recruited to account for potential dropouts. Quantitative variables (i.e., demographics), if normally distributed, were described as mean ± standard deviation (SD); otherwise, the median, minimum, maximum, and interquartile ranges were shown. To evaluate changes over time before and after the treatment, paired t-tests for dependent variables and one-way analysis of variance were performed. Statistical significance was set at *p* < 0.05. Statistical analyses were performed using SPSS Statistics V21.0 (IBM Corporation, Armonk, NY, USA).

## 3. Results

A total of 85 patients were recruited in this study. Five patients did not meet the inclusion criteria. Two participants, one for each group, dropped out at the 8th week for personal reasons unrelated to the study; thus, 78 subjects, 39 in each group, completed the trial (Figure 1).

No significant difference in any characteristic was found at baseline between the groups (Table 1 and Table 2). After 120 days of treatment, patients in Groups 1 and 2 received a mean of 46.9 and 48.4 TENS sessions, respectively.

The between-group difference in VAS scores for burning/pain and dyspareunia was significantly lower after TENS treatment (Table 3). By day 120, there was a 38.2% cumulative reduction in the burning/pain VAS scores observed in Group 1, as compared to 21.3% in Group 2 (*p* = 0.003). Similarly, there was a 52.1% cumulative reduction in the dyspareunia VAS scores for Group 1, as compared to 23.1% in Group 2 (*p* = 0.01) (Table 3). For V-Q, there was a reduction in the score in both groups without statistical significance, although there were differences between the groups after treatment, with a percentage reduction of approximately 50% in Group 1 compared to Group 2 (Group 1: −28.2%; Group 2: −13.8%, not significant) (Table 3). The results of the FSFI total score showed an improvement in both groups, with a slight advantage of 8.8% in Group 1 (Group 1: +22.9%; Group 2: +14.1%, not significant), but without a significant difference after treatment (Table 3).

Comparing data analysis 60- and 120-days post-treatment, the severity score of burning/pain for Group 1 significantly decreased to −17.6% from baseline by day 60, continuing to decrease through treatment before reaching −38.2% at day 120 (*p* = 0.003). Conversely, in Group 2, the decrease 60- and 120-days post-treatment was not statistically significant (60 days = −12.35%; 120 days = −21.33%, not significant) (Figure 2). Similarly, dyspareunia gradually decreased from 33.1% at day 60 to 52.1% at day 120 for Group 1 (*p* = 0.03), while Group 2 exhibited a small difference between 60- and 120-days post-treatment (60 days = −20.0%; 120 days = −23.7%, not significant) (Figure 2). Both groups also showed a significant reduction in cotton swab test scores at the end of treatment (Group 1: −44.2%; Group 2: −32.9%, *p* = 0.01) (Table 3). The intergroup analysis of muscle strength (Tmax-Tmin) after the treatment showed no statistical differences, despite an improvement in both groups (Group 1: +23.01%; Group 2: +17.08%, not significant) (Table 3). No adverse effects occurred in any group during the study period.

## 4. Discussion

Our results showed the effectiveness of self-applied TENS in the treatment of VBD, with a significant reduction in burning/pain and dyspareunia. In addition, the average FSFI total and V-Q questionnaire scores were significantly lower after treatment. Furthermore, the improvement was more pronounced in the group of patients who underwent specific parameters of stimulation (frequency and pulse duration), according to our previous studies [12,13] and clinical trials of TENS in relieving neuropathic pain [19].

As a non-invasive and non-pharmacological treatment, TENS has been used to treat many types of neuropathic pain [20]. The analgesic effect of TENS is achieved through different neurobiological mechanisms affecting the peripheral and central nervous systems.

A different pulse duration in high- and low-frequency TENS could be a decisive factor for efficacy. Several studies involving TENS in neuropathic pain reported that the combination of high frequency (100 Hz or more) and short pulse duration (50–60 µs) works through a spinal effect that is related to ‘pain gate’ mechanisms, in which the stimulation of large-diameter Aβ non-noxious afferent fibers (conveying activity related to touch perception, etc.) inhibits central nociceptive transmission, consequently decreasing pain perception [21]. In contrast, the use of a low frequency (5 Hz) and long pulse duration (100 µs) mainly stimulates small nociceptive fibers (Aδ and C), resulting in the release of endogenous opioids (suprasegmental effect). This is a possible reason for the results relative to the choice of both high- and low-frequency TENS in the same session in Group 1.

Furthermore, there was an increase in β-endorphins in the bloodstream and cerebrospinal fluid in healthy subjects after administration of either high- or low-frequency TENS [22]. High- and low-frequency TENS activates μ- and δ-opioid receptors in the spinal cord and brainstem, respectively [22].

It has been reported that repeated use of TENS at equal parameters (frequency and pulse duration) can produce analgesic tolerance, which may decrease its positive effects. In contrast, a modulated TENS that applies stimulation across a range of frequencies may prevent the development of tolerance to electrical stimulation [23]. As with the previous studies, we demonstrated that TENS gradually decreased symptoms related to VBD from days 60 to 120 without any declining effect over time. In addition, TENS was more effective in Group 1, in which the parameters were more modulated than in Group 2.

Most women with VBD described their pain as “hot”, “burning”, or “pricking”, that the vestibular area was sensitive to touch (e.g., during sexual intercourse or tampon use), and that the pain would be aggravated by rubbing [24].

The pattern of VBD responses suggests sensory abnormalities in the form of evoked pain (e.g., hyperalgesia or allodynia), suggesting sensitization as a manifestation of neuropathic pain [25]. Furthermore, the discomfort in VBD is always associated with pelvic floor muscle overactivity [9]. This prolonged pattern can result in decreased tissue perfusion, muscle overactivity, and the development of myofascial trigger points, resulting in localized or radiating pain and/or intense tenderness. Neuropathic pain and hypertonicity can be considered as a multifactorial and complex consequence of maladaptive neuronal plasticity. TENS is thought to address this by activating a complex neuronal network leading to a reduction in pain [26]. In our study, we directed TENS to various abnormal vestibular sensory nerve fiber anomalies (e.g., hyperalgesia and allodynia) that affected patients with VBD, based on the stimulation parameters and the site of application (vulvar vestibule). We also observed an improvement in muscle strength (Tmax-Tmin) after the treatment in both groups, despite the lack of statistical differences (Group 1: +23.01%; Group 2: +17.08%, not significant). This is a pivotal parameter because introital dyspareunia is related to overactivity and a decrease in the strength and endurance of the pelvic floor muscles [27]. We can speculate that TENS can help reduce PFM dysfunction because it reduces vulvar pain, resulting in a secondary decrease in muscle activity and spasm. In addition, an overlap was observed between the dendrites of the levator ani motor neurons and pudendal neurons, with an interaction between the sensory and motor nerve fibers of the PFM and vulvar vestibule [28]. Our results regarding the use of TENS for VBD-related symptoms were comparable with those reported by Vallinga et al. [14] in the context of administering TENS as a domiciliary protocol. However, in general, our study cannot be compared with the study by Vallinga et al. [14] because of the differences in treatment protocols. While we used specific stimulation parameters (frequency and pulse duration), they adopted stimulation parameters that were not standardized for each patient. The frequency was set at 80 Hz with a pulse duration of 50 or 180 µs, according to a woman’s perception. Furthermore, the TENS treatment was administered at home two to three times a day for a total of 90 min per day, three times per week. The authors did not report any data on pelvic floor muscle strength and endurance.

A limitation of this study is the lack of a placebo and control group and follow-up without treatment, although placebo TENS was already compared with functional TENS in our previous randomized controlled trial on women with VBD with three months follow-up [13]. Our findings demonstrate the effectiveness of TENS in the treatment of women with VBD. It has the advantages of being easy to use, safe, portable, and inexpensive. Self-management can resolve compliance issues related to treatment duration and frequency and allows participants to monitor progress by encouraging adherence to treatment.

It is becoming increasingly apparent that VBD is a summation and overlap of various trigger factors (infections, hormonal disturbances, allergies, genetic aspects, psychological vulnerability, and others) with weight and predominance varying from patient to patient, in which the common endpoint is vestibular hypersensitivity and pelvic floor hypertonic dysfunction. As a result, any single-approach management strategy may not be adequate for all women. TENS is usually administered to patients as an adjunct treatment for pain reduction and is not given as a sole treatment.

## 5. Conclusions

Our findings confirm that TENS therapy is effective for reducing pain and sexual problems in patients with VBD. Therefore, we recommend that TENS be administered not as a single therapy but as an important part of multidisciplinary therapy, in order to maximize its effectiveness in the treatment of VBD. However, proper stimulation parameters must be identified.

## Figures and Tables

**Figure 1 medsci-11-00048-f001:**
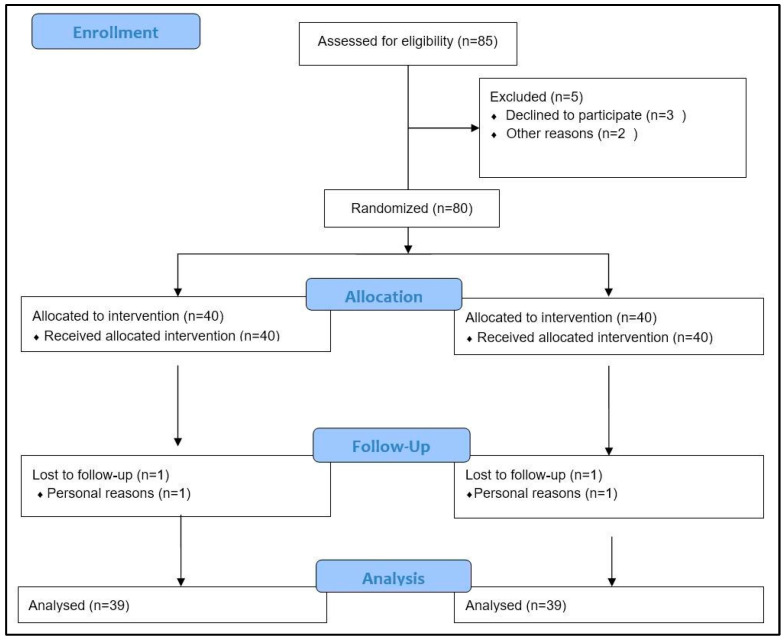
Study flow chart.

**Figure 2 medsci-11-00048-f002:**
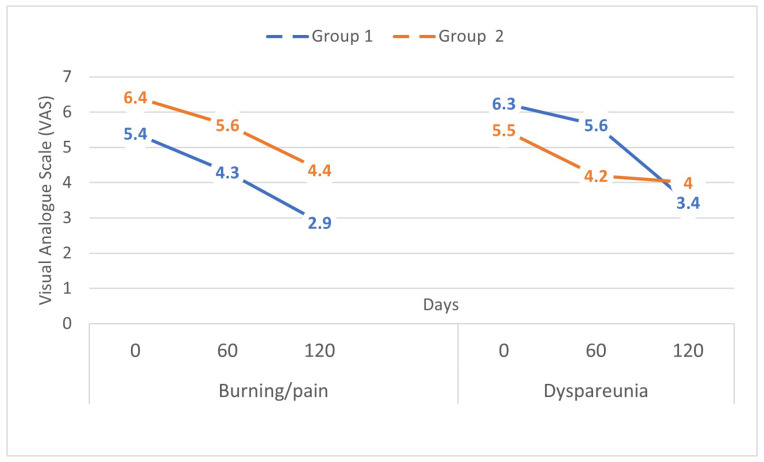
Burning/pain and dyspareunia changes after 60 and 120 days of treatment.

**Table 1 medsci-11-00048-t001:** Demographics and baseline characteristics.

	Total Sample n = 78	Group 1 n = 39	Group 2 n = 39	*p* Value
Age, years *	27.3 (6.3)	27.0 (1.2)	27.7 (2.0)	NS
Duration of disease (months *)	31.8 (24.5)	39.0 (28.5)	29.5 (17.5)	NS
Use of oral contraceptives	31	15	16	NS
Duration (months *)	41.0 (41.6)	35.3 (33.2)	46.7 (49)	
Menstruation range (days)				
25 to less than 35	60	28	32	NS
Less than 25 gg	7	4	3	NS
35 or more	11	7	4	NS
Previous therapies °				
-Tricyclic antidepressant	10	6	4	NS
-Desensitizing gel	13	8	5	NS
-EMG biofeedback	9	5	4	NS
-None	46	20	26	NS

* mean (SD), ° number. NS—Not significant.

**Table 2 medsci-11-00048-t002:** Distribution of symptoms and vaginal EMG measurements at baseline.

	Group 1 (n = 39)	Group 2 (n = 39)	*p* Value
Burning/pain (VAS) *	5.4 (0.6)	6.4 (0.6)	NS
Dyspareunia score (VAS) *	6.2 (0.7)	5.5 (0.8)	NS
V-Q questionnaire	13.6 (1.3)	12.4 (1.1)	NS
FSFI	15.6 (2.0)	16.9 (1.7)	NS
Cotton swab test *	6.3 (0.4)	6.7 (0.4)	NS
Vaginal EMG measurements (μV) *			NS
-tone value at rest (Tmin)	3.2 (0.3)	3.6 (0.3)	
-maximal PFM contraction (Tmax)	16.6 (1.3)	15.9 (1.4)	
-PFM strength (Tmax-Tmin)	13.4 (1.3)	12.2 (1.5)	

* mean (SD). Abbreviation: Visual analogue scale (VAS); EMG—electromyography; PFM—pelvic floor muscle; μV—micronvolt; NS—not significant.

**Table 3 medsci-11-00048-t003:** Change of symptoms and signs at baseline and after 120 days.

KERRYPNX	Group 1 (n = 39)		*p*-Value	Group 2 (n = 39)		*p*-Value
	Baseline	after 120 Days		Baseline	after 120 Days	
Burning/pain (VAS) *	5.4 (0.6)	2.8 (0.7)	0.003	6.4 (0.6)	4.3 (0.6)	0.01
Dyspareunia score (VAS) *	6.2 (0.7)	3.4 (0.9)	0.01	5.5 (0.8)	4.0 (0.7)	NS
V-Q questionnaire	13.6 (1.3)	9.8 (1.3)	NS	12.4 (1.1)	10.2 (1.1)	NS
FSFI	15.6 (2.0)	20.2 (2.7)	NS	16.9 (1.7)	19.7 (2.1)	NS
Cotton swab test *	6.3 (0.4)	3.3 (0.7)	0.01	6.7 (0.4)	4.1 (0.7)	NS
Vaginal EMG measurements (μV) *						NS
-tone value at rest (Tmin)	3.2 (0.3)	2.7 (0.2)		3.6(0.3)	2.8 (0.3)	
-maximal PFM contraction (Tmax)	16.6 (1.3)	19.3 (2.4)		15.9 (1.4)	16.0 (1.8)	
-PFM strength (Tmax-Tmin)	13.4 (1.3)	16.6 (2.3)		12.2(1.5)	13.2 (1.8)	

* mean (SD). Abbreviation: Visual analogue scale (VAS); EMG—electromyography; PFM—pelvic floor muscle; μV—micronvolt; NS—not significant mean.

## Data Availability

Data is unavailable due to privacy and institutional restrictions.

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
