# Peer review of "Effectiveness of Two Transcutaneous Electrical Nerve Stimulation (TENS) Protocols in Women with Provoked Vestibulodynia: A Randomized Controlled Trial"

_medsci, 2023, doi:10.3390/medsci11030048_

Round 1

Reviewer 1 Report

Dear Authors,

Thank you for giving me the opportunity to review your article titled " Effectiveness of two Transcutaneous Electrical Nerve Stimulation (TENS) protocols in women with provoked vestibulodynia: a randomized controlled trial”. After conducting a thorough review, I have come to the conclusion that the topic addressed in your article is of high interest to the scientific community. However, there are some issues that need to be addressed before it can be considered for publication.

Firstly, there are some language issues that need to be revised. Some sentences are not grammatically correct and there are some typographical errors that need to be corrected. Additionally, the images in the article are pixelated and illegible, which detracts from the overall quality of the paper.

Secondly, while the intervention you describe in the article - using transcutaneous electrical nerve stimulation (TENS) at different frequencies - is interesting and novel, there are methodological flaws that make it difficult to validate your results and draw conclusions from your research. Specifically, the lack of a control group makes the methodology inadequate for drawing inferences about the treatment's effectiveness. Additionally, all participants received the treatment, which undermines the possibility of conducting a double-blind study, as you described.

Considering the importance of the topic, I understand that it may be difficult to address these issues in a timely and comprehensive manner.

I hope that my feedback will be helpful in guiding your future research efforts, and I encourage you to continue exploring this important topic with the highest standards of scientific rigor.

Thank you again for considering my review, and I wish you all the best in your future endeavors.

Sincerely,

There are some language issues that need to be revised. Some sentences are not grammatically correct and there are some typographical errors that need to be corrected. 

Author Response

-Reviewer 1:  

Firstly, there are some language issues that need to be revised. Some sentences are not grammatically correct and there are some typographical errors that need to be corrected. Additionally, the images in the article are pixelated and illegible, which detracts from the overall quality of the paper. 

Author’s response and action taken.

We revised the language and the grammar of the paper, and we modified the images, as suggested by reviewer.

Secondly, while the intervention you describe in the article - using transcutaneous electrical nerve stimulation (TENS) at different frequencies - is interesting and novel, there are methodological flaws that make it difficult to validate your results and draw conclusions from your research. Specifically, the lack of a control group makes the methodology inadequate for drawing inferences about the treatment's effectiveness. Additionally, all participants received the treatment, which undermines the possibility of conducting a double-blind study, as you described. Considering the importance of the topic, I understand that it may be difficult to address these issues in a timely and comprehensive manner. I hope that my feedback will be helpful in guiding your future research efforts, and I encourage you to continue exploring this important topic with the highest standards of scientific rigor. 

Author’s response

Reviewer’s comment is very pertinent. Of course, we understand that our paper lack of a control group, but we demonstrated in our previous study the effectiveness of TENS in a randomized trial (Murina F, Bianco V, Radici G, Felice R, Di Martino M, Nicolini U. Transcutaneous electrical nerve stimulation to treat vestibulodynia: a randomised controlled trial. BJOG. 2008 Aug;115(9):1165-70.). The goal of this research was the demonstration of proper TENS parameters to obtain best results in the treatment of vestibulodynia.

Reviewer 2 Report

Dear authors,

It was a pleasure for me to review this manuscript that deals with a randomized controlled trial to verify the efficacy of two transcutaneous electrical stimulation protocols in women with vetibulodynia.

The topic of the study seemed very novel and interesting to me.

The intro section seemed fine to me. The authors clearly explained each of the concepts, and the reader is immediately placed in the context of the situation required in the latest published evidence in this regard. At the end of this section, the objective of the research was clearly defined.

As for the material section, I have to say that it is also well structured, it perfectly explains how the groups were made and how the sample was selected to be representative. In my opinion, I think the methodology is well established.

The results are also clearly indicated.

The discussion also seemed correct to me, discussing the results obtained with other studies and acknowledging the limitations of the study.

The conclusions section seemed very short to me, I think it does not adequately support the very interesting results that were found. To put a hitch, I would suggest rewriting the conclusions obtained in the results a little more. I would have liked to read in this section which of the two protocols is the best and why.

For the rest, I would like to congratulate the authors for the extremely interesting work they have developed.

Thank you

Kind regards

Author Response

Reviewer 2.

The conclusions section seemed very short to me, I think it does not adequately support the very interesting results that were found. To put a hitch, I would suggest rewriting the conclusions obtained in the results a little more. I would have liked to read in this section which of the two protocols is the best and why.

Author’s response and action taken.

We improved the conclusion, as suggested by reviewer.

Round 2

Reviewer 1 Report

I have thoroughly reviewed your submitted study, and I greatly appreciate the effort and research conducted. However, I believe that several major changes are required to strengthen the study's overall quality and address certain limitations. Additionally, we would like to acknowledge the study's interesting findings. Please find the detailed feedback below:

Format Issue: Tables 1 and 2:

We kindly request that you revise the information presented between Tables 1 and 2, as it appears to be unclear and improperly placed. Ensuring the correct format and placement of these tables is essential to enhance the overall readability and organization of the document. Please make the necessary adjustments to rectify this issue.

Statistical Analysis:

On Line 144 of the manuscript, it is mentioned that the statistical analysis was conducted using SPSS 21. However, the correct version cited should be SPSS Statistics V21.0. We kindly ask you to rectify this error by stating the accurate software version used in the analysis.

Limitations:

In Line 303, you mentioned a limitation of the study, stating, "A limitation of this study is the lack of a placebo group and follow-up without treatment, although placebo TENS was already compared with functional TENS in our previous randomized controlled trial on women with VBD with three months follow-up [14]." We recommend revising this sentence to specify "placebo or control group" instead of solely mentioning "placebo group." This modification will provide a more comprehensive understanding of the study's limitations.

Furthermore, in Line 64, it is stated that the study cannot be considered a double-blind study. We suggest amending this to "single blind" to accurately represent the study design. As I understand only the assessor is blinded, not the interventor neither the participant (even if the participant would not know which is the frequency applied, if the participant had already received previously a TENS treatment she could recognise). Could you explain it better?

Additionally, we acknowledge your findings regarding the improvement of pain by more than 2 points on the Visual Analog Scale (VAS). Although the results may not be statistically significant, they hold clinical relevance. We encourage you to include this information in the text to highlight the clinical significance of these findings.  The clinical relevance of the treatment should be considered as an effect on this profile. VAS value reduction of more than 2 points is clinically relevant.

Finally, we recommend addressing the potential bias arising from participants who may have previously received TENs treatment. Registering whether participants had prior experience with TENs and considering their expectations as a potential bias should be added to the study's limitations section. This addition will contribute to a more thorough analysis and interpretation of the study's results.

We understand the significance of your study and find it to be of great interest to the scientific community. Therefore, it is crucial to ensure that the necessary changes are implemented to enhance the study's reliability, validity, and overall contribution to the field.

Thank you for your attention to these matters. We look forward to receiving the revised manuscript addressing the mentioned concerns and eagerly anticipate the opportunity to re-evaluate your study.

 Sincerely,

Author Response

Format Issue: Tables 1 and 2:

We kindly request that you revise the information presented between Tables 1 and 2, as it appears to be unclear and improperly placed. Ensuring the correct format and placement of these tables is essential to enhance the overall readability and organization of the document. Please make the necessary adjustments to rectify this issue.

Autor’s response and action taken.

We revised the tables, as suggested by reviewer.

Statistical Analysis:

On Line 144 of the manuscript, it is mentioned that the statistical analysis was conducted using SPSS 21. However, the correct version cited should be SPSS Statistics V21.0. We kindly ask you to rectify this error by stating the accurate software version used in the analysis.

Autor’s response and action taken.

We modified the specification of software , as suggested by reviewer.

Limitations:

In Line 303, you mentioned a limitation of the study, stating, "A limitation of this study is the lack of a placebo group and follow-up without treatment, although placebo TENS was already compared with functional TENS in our previous randomized controlled trial on women with VBD with three months follow-up [14]." We recommend revising this sentence to specify "placebo or control group" instead of solely mentioning "placebo group." This modification will provide a more comprehensive understanding of the study's limitations.

Autor’s response and action taken.

We inserted the term “control”, as suggested by reviewer.

Furthermore, in Line 64, it is stated that the study cannot be considered a double-blind study. We suggest amending this to "single blind" to accurately represent the study design. As I understand only the assessor is blinded, not the interventor neither the participant (even if the participant would not know which is the frequency applied, if the participant had already received previously a TENS treatment she could recognise). Could you explain it better?

Autor’s response and action taken.

We modified the term, as suggested by reviewer. However, one or more members of the staff who do not work directly with the subject will be responsible for device settings and assignment to one of two treatments based on random assignment. Patients were naïve for TENS treatment, so that they were unable to distinguish the differences.

Additionally, we acknowledge your findings regarding the improvement of pain by more than 2 points on the Visual Analog Scale (VAS). Although the results may not be statistically significant, they hold clinical relevance. We encourage you to include this information in the text to highlight the clinical significance of these findings.  The clinical relevance of the treatment should be considered as an effect on this profile. VAS value reduction of more than 2 points is clinically relevant.

Autor’s response and action taken.

We inserted a new table (Table 3) to better explain our results.

Finally, we recommend addressing the potential bias arising from participants who may have previously received TENs treatment. Registering whether participants had prior experience with TENs and considering their expectations as a potential bias should be added to the study's limitations section. This addition will contribute to a more thorough analysis and interpretation of the study's results.

Autor’s response and action taken.

All patients included in the trial never did before Tens therapy. We added this specification in exclusion criteria.